# Efficacy and Safety of Preoperative Melatonin for Women Undergoing Cesarean Section: A Systematic Review and Meta-Analysis of Randomized Placebo-Controlled Trials

**DOI:** 10.3390/medicina59061065

**Published:** 2023-06-01

**Authors:** Wardah Albzea, Lolwa Almonayea, Marah Aljassar, Mousa Atmeh, Khaled Al Sadder, Yousef AlQattan, Raghad Alhajaji, Hiba AlNadwi, Inaam Alnami, Fatima Alhajaji

**Affiliations:** 1Faculty of Medicine, Alexandria University, Alexandria 21544, Egypt; 2Kuwait Institute for Medical Specializations, Kuwait City 12050, Kuwait; 3Department of Hemto-Oncology, Royal Medical Services, Amman 11855, Jordan; 4Department of General Surgery, Ministry of Health, Kuwait City 12009, Kuwait; 5Department of Family Medicine, Almagrah Primary Health Care, Ministry of Health, Makkah 11176, Saudi Arabia; 6King Abdullah Medical City, Makkah 57657, Saudi Arabia; 7Senior Registerar Family Medicine, Internal Medicine Department, Security Forces Hospital Program, Makkah 14799, Saudi Arabia; 8College of Medicine, Umm Alqura University, Makkah 57483, Saudi Arabia

**Keywords:** cesarean section, melatonin, pain, blood loss

## Abstract

*Background*: Cesarean section (CS) has been linked to a number of negative effects, such as pain, anxiety, and sleeping problems. The aim of this systematic review and meta-analysis was to investigate the safety and efficacy of preoperative melatonin on postoperative outcomes in pregnant women who were scheduled for elective CS. *Methods*: We systemically searched 4 electronic databases (PubMed, Scopus, Web of Science, and Cochrane Library) from inception until 10 March 2023. We included randomized controlled trials (RCTs) comparing melatonin and placebo for postoperative outcomes in CS patients. For risk of bias assessment, we used the Cochrane Risk of Bias 2 tool. Continuous variables were pooled as mean difference (MD), and categorical variables were pooled as a risk ratio (RR) with a 95% confidence interval (CI). *Results*: We included 7 studies with a total of 754 pregnant women scheduled for CS. The melatonin group had a lower pain score (MD = −1.23, 95% CI [−1.94, −0.51], *p* < 0.001) and longer time to first analgesic request (MD = 60.41 min, 95% CI [45.47, 75.36], *p* < 0.001) than the placebo group. No difference was found regarding hemoglobin levels, heart rate, mean arterial pressure, total blood loss, or adverse events. *Conclusions*: Preoperative melatonin may reduce postoperative pain in CS patients without side effects. This research offers a safe and affordable pain management method for this population, which has clinical consequences. Further research is needed to validate these findings and determine the best melatonin dosage and timing.

## 1. Introduction

Cesarean section (CS) is one of the most popular surgical procedures performed worldwide, and its prevalence has progressively increased over the last few decades [1,2,3,4]. It is usually carried out when a vaginal birth cannot be accomplished or is deemed dangerous for the mother or the infant [5,6]. In many situations, including fetal distress, breech presentation, placenta previa, and maternal health problems such as hypertension, diabetes, or infections that might make vaginal birth problematic for the mother or baby, CS may be lifesaving for both the mother and fetus [7,8,9]. Although CS has the potential to save lives in certain circumstances, there are dangers associated with the surgery, such as hemorrhage, infection, organ damage, and a longer recovery period than with vaginal birth [9,10,11,12]. Moreover, CS has been linked to a number of negative effects, such as pain, anxiety, and sleeping problems [13,14,15]. Many pharmaceutical therapies have been recommended to reduce these adverse effects and enhance mother and fetal outcomes, including the use of melatonin [16,17].

Melatonin is a hormone generated by the pineal gland, a tiny endocrine gland situated in the brain that regulates the circadian rhythm [18,19]. The circadian rhythm is the body’s internal clock, which includes the sleep–wake cycle and other physiological processes that follow a 24 h cycle [20]. Melatonin is normally produced at night in response to darkness, reaching its peak between 2 and 4 am and then progressively decreasing during the day [21]. In addition to its sleep-promoting benefits, melatonin has been proven to have anti-inflammatory, antioxidant, and analgesic properties, making it a potentially helpful supplement for perioperative treatment [22,23,24,25]. Melatonin has been shown to improve postoperative outcomes by lowering postoperative pain, anxiety, and sleep disruptions, according to recent meta-analyses [26,27,28]. Melatonin has been investigated in recent studies as a preoperative intervention to enhance CS results [16,29,30], although the evidence is still limited and contradictory.

Therefore, the aim of this systematic review and meta-analysis of randomized controlled trials (RCTs) was to investigate the safety and efficacy of preoperative melatonin on postoperative outcomes in pregnant women who were scheduled for elective CS.

## 2. Methods

We followed the principles outlined in the PRISMA declaration while conducting our systematic review and meta-analysis [31]. The Cochrane Handbook of Systematic Reviews and Meta-analysis of Interventions (version 6.2) was strictly followed at every step [32]. In addition, this review was registered in the PROSPERO database (registration ID: CRD42022363653).

### 2.1. Eligibility Criteria

Studies that met the following criteria were included in our analysis: Population: women who had undergone CS. Intervention: melatonin. Comparator: studies with placebo or no treatment. Outcomes: studies reporting at least one of the following outcomes: hemoglobin, heart rate, mean arterial pressure, anxiety score, pain score, time to first analgesic request, total blood loss, and adverse events. Study design: Comparative studies using the design of controlled trials in which patients were randomly assigned to either melatonin or placebo. Both blinded and open-label trials were evaluated. We excluded studies whose data were not reliable for extraction and analysis, observational studies, case reports, case series, and review articles.

### 2.2. Information Sources and Search Strategy

We performed a comprehensive search of 4 electronic databases (PubMed, Scopus, Web of Science, and Cochrane Library) from inception until 10 March 2023 using the following query: (Melatonin OR pineal hormone OR NSC-113928) AND (Cesarean section OR Caesarean Section OR Abdominal Deliveries OR Abdominal delivery OR C-Section OR C Section OR Postcesarean Section). In addition, the references of the included papers were manually screened for any potential studies. The details of the search strategy for each database are presented in Appendix A.

### 2.3. Selection Process

Endnote (Clarivate Analytics, Philadelphia, PA, USA) was used to eliminate duplicates, and the obtained references were screened in two phases: the first step consisted of screening the titles/abstracts of all selected articles independently by all authors to determine their relevance to this meta-analysis, and the second step consisted of screening the full-text articles of the included abstracts for final eligibility for the meta-analysis.

### 2.4. Data Collection Process and Data Items

Data were extracted to a uniform data extraction sheet. The extracted data included (1) characteristics of the included studies (study ID, design, country, interventions, sample size, time of administration, administration route, time of anesthesia, and follow-up period), (2) characteristics of the population of included studies (age, weight, duration of surgery, and preoperative hemoglobin), (3) risk of bias domains, and (4) outcome measures (postoperative hemoglobin, heart rate, mean arterial pressure, anxiety score, 10-point pain score, time to first analgesic request, total blood loss, and adverse events).

### 2.5. Assessing the Risk of Bias in the Individual Studies

We used the Cochrane assessment tool 2 (ROB2) for RCTs [33]. The risk of bias assessment included the following domains: bias arising from the randomization process, bias due to deviations from intended interventions, bias due to missing outcome data, bias in the measurement of the outcome, bias in the selection of the reported result, and other bias. The authors’ judgments were categorized as “low risk”, “high risk”, or “some concerns” of bias.

### 2.6. Synthesis Methods

The analysis was performed using Review Manager Software (RevMan 5.4.1) under the inverse variance method. The continuous variables were pooled as mean difference (MD), and the dichotomous variables were pooled as risk ratio (RR) in a random-effect model with a relative 95% CI. A *p*-value < 0.05 was considered statistically significant.

### 2.7. Choice of the Meta-Analysis Model

Using the DerSimonian–Liard meta-analysis model [32], we estimated the pooled effect size for all outcomes. This random-effect model implies that the included studies reflect a random sample of the population and allocates somewhat more weight to small studies relative to the costs of larger studies. We selected this model because, in contrast to the fixed-effects model, it allows for a higher standard error in the pooled estimate, making it appropriate in the event of inconsistent or debatable estimates. Thus, the predicted effects in our meta-analysis are cautious estimates that account for the possibility of discrepancies.

### 2.8. Assessment of Heterogeneity

Statistical heterogeneity among the studies was evaluated by the chi-square test (Cochrane Q test) [32]. Then, the chi-square statistic, Cochrane Q, was used to calculate the I-squared according to the equation: I^2^ = $\left(\frac{Q-df}{Q}\right)\times 100\%$. A chi-square *p*-value less than 0.1 was considered as significant heterogeneity. I-squared values ≥50% were indicative of high heterogeneity.

### 2.9. Reporting Bias Assessment

According to Egger and colleagues [34,35], publication bias evaluation is inaccurate for 10 pooled studies; therefore, we were unable to detect the presence of publication bias in the current research using Egger’s test for funnel plot asymmetry. Moreover, we used the Grading for Recommendation Assessment, Development, and Evolution (GRADE) method to grade the overall certainty of evidence. 

## 3. Results

### 3.1. Literature Search Results

Our search for relevant literature yielded 382 results. After screening for titles and abstracts, 22 papers qualified for full-text review. Of them, five were included in the meta-analysis. In addition, two studies were included after manually searching the references of the included studies. So, finally, seven studies were included. Figure 1 depicts the PRISMA flowchart of the research selection procedure.

### 3.2. Characteristics of the Included Studies

We included 7 studies in the meta-analysis with a total of 754 pregnant women scheduled for CS. In all studies, patients were assigned to receive either melatonin or placebo. A summary and baseline of the characteristics of the included studies and their population are provided in Table 1 and Table 2. According to the Cochrane ROB tool 2, two studies had a high risk of bias, two studies had some concerns of bias, and three studies had a low risk of bias (Figure 2). Regarding the randomization process domain, three RCTs [36,37,38] were evaluated as having “some concerns” due to missing information about the randomization process. For the deviation from intended interventions domain, two RCTs [36,38] were evaluated as having “some concerns” due to no information about the deviations from usual practice that were likely to impact the outcome. For the missing data domain, one RCT [30] was evaluated as having a “high” risk of bias due to a high degree of missing data. For the measurement of outcomes domains, two RCTs [37,38] were evaluated as having some concerns, and one RCT [36] was evaluated as having a “high” risk of bias. Moreover, for the selection of the reported results domain, four RCTs [30,36,37,38] were evaluated as having “some concerns” due to no information available to exclude the possibility that reported outcome data were selected.

### 3.3. Hemoglobin (mg/dL)

The overall MD for hemoglobin levels did not favor either the melatonin or placebo groups (MD = 0.13 g/dL, 95% CI [−0.09, 0.34], *p* = 0.25, very low certainty of evidence) (Figure 3A). The pooled studies were not homogenous (*p* = 0.20; I^2^ = 37%). The subgrouping according to dose also did not favor either the melatonin or placebo groups (Table 3, Appendix A).

### 3.4. Heart Rate (bpm)

The overall MD for heart rate did not favor either the melatonin or placebo groups (MD = −2.05 beat/min, 95% CI [−7.78, 3.68], *p* = 0.48, very low certainty of evidence) (Figure 3B). The pooled studies were homogenous (*p* = 0.31; I^2^ = 0%). The subgrouping according to dose also did not favor either the melatonin or placebo groups (Table 3, Appendix A).

### 3.5. Mean Arterial Pressure

The overall MD for mean arterial pressure did not favor either the melatonin or placebo groups (MD = −2.05 mmHg, 95% CI [−9.89, 5.79], *p* = 0.61, very low certainty of evidence) (Figure 3C). The pooled studies were homogenous (*p* = 0.28; I^2^ = 16%). The subgrouping according to dose also did not favor either the melatonin or placebo groups (Table 3 and Appendix A).

### 3.6. Pain Score (10-Point)

The overall MD for pain score showed that the melatonin group had a lower pain score than the placebo group (MD = −1.23, 95% CI [−1.94, −0.51], *p* < 0.001, very low certainty of evidence) (Figure 4). The pooled studies were not homogenous (*p* < 0.001; I^2^ = 96%). The subgrouping according to the duration showed that the melatonin group had a lower pain score than the placebo group at 6 h postoperatively (MD = −1.81, 95% CI [−2.65, −0.96], *p* < 0.001) and at 12 h postoperatively (MD = −1.59, 95% CI [−2.85, −0.33], *p* = 0.01) but not at 24 h postoperatively (MD = −0.29, 95% CI [−0.59, 0.02], *p* = 0.07). The pooled studies were not homogenous for the 6 and 12 h postoperation subgroups (*p* = 0.006; I^2^ = 87% and *p* < 0.001; I^2^ = 95%, respectively) but were homogenous for the 24 h postoperation subgroup (*p* = 0.12; I^2^ = 59%). The subgrouping according to dose and duration also favored the melatonin group for all melatonin doses at 6 h and 12 h postoperatively, but at 24 h postoperatively the only dose that showed a significant lower pain score was 10 mg (Table 3, Appendix A).

### 3.7. Time to First Analgesic Request

The overall MD for time to first analgesic request showed that all melatonin subgroups had a longer time to the first analgesic request than the placebo (MD = 60.41 min, 95% CI [45.47, 75.36], *p* < 0.001, very low certainty of evidence) (Figure 5). The pooled studies were homogenous (*p* > 0.71; I^2^ = 0%). The subgrouping according to dose showed that melatonin doses 1.5, 3, 5, and 10 mg, but not 6 mg, had a longer time to the first analgesic request than the placebo (Table 3, Appendix A). The longest time was for the 10 mg subgroup (MD = 90.00 min, 95% CI [57.25, 122.75], *p* < 0.001).

### 3.8. Total Blood Loss

The overall MD for total blood loss did not favor either the melatonin or placebo groups (MD = 17.16 mL, 95% CI [−20.72, 55.04], *p* = 0.37, very low certainty of evidence) (Figure 6). The pooled studies were homogenous (*p* = 0.21; I^2^ = 37%). The subgrouping according to dose also did not favor either the melatonin or placebo groups (Table 3, Appendix A).

### 3.9. Adverse Events

The pooled RR for the total adverse events did not favor either of the two groups; the use of melatonin in this population did not significantly increase the risk of adverse events (RR = 1.02, 95% CI [0.73, 1.42], *p* = 0.92, very low certainty of evidence) (Figure 7 and Appendix A). The pooled studies were homogenous (*p* = 0.18; I^2^ = 35%). The subgrouping according to the type of the adverse event reported showed that the use of melatonin did not increase the risk of nausea (RR = 0.91, 95% CI [0.57, 1.46], *p* = 0.70), vomiting (RR = 0.94, 95% CI [0.46, 1.95], *p* = 0.88), or headache (RR = 1.48, 95% CI [0.60, 3.63], *p* = 0.39). The pooled studies were homogenous for all subgroups (*p* > 0.1; I^2^ < 50%). The subgrouping according to dose for each adverse event reported showed that none of the melatonin doses increased the risk of any adverse event except the 6 mg melatonin subgroup, which significantly increased the risk of headache (RR = 4.28 [1.59, 11.48], *p* = 0.004) (Table 3 and Appendix A).

## 4. Discussion

### 4.1. Significance of the Study

To the best of our knowledge, this is the first systematic review and meta-analysis evaluating the safety and efficacy of preoperative melatonin in women who have undergone CS. This study helps to clarify the current evidence about the exact efficacy of preoperative melatonin on different intra- and postoperative outcomes.

### 4.2. Summary of Findings

This study found that melatonin did not have a significant effect on hemoglobin levels, heart rate, mean arterial pressure, or total blood loss compared with the placebo. However, patients who received melatonin had a lower pain score and longer time before the first analgesic request compared with those who received the placebo. Melatonin did not increase the risk of adverse events. The subgroup analysis based on dose and duration showed varying results for the pain score and time to first analgesic request with the highest efficacy associated with the higher doses of melatonin.

### 4.3. Explanation of the Findings

In recent years, many studies have been conducted aiming to investigate and discover the different and numerous effects of melatonin [40]. We found that melatonin significantly reduces postoperative pain and prolongs the time to first analgesic request, which can be attributed to several mechanisms. First, melatonin has been found to possess anti-inflammatory properties by suppressing the generation of pro-inflammatory cytokines and reducing the activity of inflammatory cells [41,42]. Second, melatonin modulates the perception of pain by interacting with the opioid system of the brain [43,44,45]. Opioids are a natural compound in the brain’s “endogenous opioids” and a type of medication commonly used to relieve pain. They function by attaching to certain brain receptors [46]. It has been demonstrated that melatonin increases the activity of these opioid receptors, thus reducing pain [43,47,48,49]. Third, melatonin modulates the activity of other neurotransmitters involved in pain perception, including serotonin and dopamine [43,45,50], in addition to its antioxidant and opioid-related activities, which can help relieve pain and induce relaxation. Fourth, melatonin is popularly known as a sleep aid, and research indicates that it can increase the quality and length of sleep [19,21,25,26]. Because poor sleep quality is known to worsen pain sensitivity and perception [51,52,53,54], melatonin supplementation can lower postoperative pain indirectly. Fifth, anxiety before surgery is a typical condition that can exacerbate postoperative pain [55,56,57]. Melatonin has been proven to reduce the activity of the sympathetic nervous system, which is responsible for the “fight or flight” response; therefore, it can indirectly lower postoperative pain by reducing anxiety [24,26,27,28]. Finally, melatonin’s muscle-relaxing qualities can alleviate muscular tension and spasm [58,59]. This is especially advantageous for CS, as muscular tension and spasm can add to postoperative pain. Our results support the current evidence from systematic reviews and meta-analyses on other types of surgeries about the analgesic effect of melatonin [60,61].

We found a non-significant effect of melatonin on hemoglobin levels, heart rate, mean arterial pressure, and total blood loss, which may be explained by a lack of direct influence of melatonin on these variables because heart rate and mean arterial pressure are primarily controlled by the cardiovascular system [62,63]. However, the association between melatonin and heart rate and mean arterial pressure could be explained by the effect of melatonin on the sympathetic nervous system [18]. Furthermore, hemoglobin levels and blood loss are influenced by surgical technique, patient characteristics, and the kind of operation [64,65]. In addition, the study’s sample size may not have been large enough to find statistically significant changes between these variables. It may require a larger sample size to evaluate whether melatonin has any meaningful effects on hemoglobin levels, heart rate, mean arterial pressure, or total blood loss.

Our results suggest that melatonin is a relatively safe adjuvant therapy with no difference in terms of adverse events reported such as nausea, vomiting, and headache, which agrees with the previous meta-analysis of the adverse events reported with melatonin [60,61].

### 4.4. Implications of These Findings in Practice

Our findings are important for healthcare providers as they can reassure them that the use of melatonin as an adjunct therapy does not pose a risk to the patient’s hemodynamic status or of blood loss. Furthermore, the lack of effect on hemoglobin levels suggests that melatonin does not interfere with the body’s ability to form blood clots. This is especially relevant for patients undergoing surgical procedures that carry a risk of bleeding complications. The findings indicate that preoperative melatonin does not increase the risk of bleeding or hemodynamic instability in these patients. Overall, the implications of these findings are that preoperative melatonin can be safely used as an adjunct therapy to reduce postoperative pain in patients undergoing cesarean section without adversely affecting hemoglobin levels, heart rate, mean arterial pressure, or total blood loss. This information is important for clinicians as melatonin can provide a safe and effective alternative to traditional pain management strategies.

### 4.5. Strengths and Limitations

This study has several strengths. It is the first, most comprehensive, and up-to-date meta-analysis evaluating the safety and efficacy of melatonin on women who have undergone CS. In addition, we used strict inclusion and exclusion criteria which helped to ensure that the studies included in our analysis were of high quality and were comparable to one another. Moreover, we did not apply a restriction on the language of the published articles; we included articles published in all languages and translated them to English before the data extraction. However, this meta-analysis has some limitations, including the relatively small number of included studies (only seven) with a small sample size. In addition, not all studies reported all outcomes of interest. Moreover, there was inconsistency in several outcomes, and there was a wide difference in dosing between studies.

### 4.6. Recommendations for Future Research and Clinical Practice

Further research is required to determine the ideal dose and timing of preoperative melatonin administration for effective pain relief in CS patients. Future research should employ larger sample sizes to increase the statistical power of the analysis and validate the findings of our study. Preoperative melatonin administration’s effects on additional outcomes in CS patients, including nausea, vomiting, anxiety, and sleep quality, need to be studied. Moreover, further secondary analysis such as a dose–response analysis are needed to determine the ideal dose of melatonin administration.

Clinicians are advised to consider preoperative melatonin administration as part of a multimodal strategy of pain management in CS patients, which may include nonpharmacological interventions such as acupuncture and relaxation techniques. Future research should examine the long-term benefits of preoperative melatonin administration for pain and other outcomes in CS patients, as well as the cost-effectiveness of this intervention.

### 4.7. Conclusions

In conclusion, our study demonstrates that preoperative administration of melatonin can be a safe and effective method for lowering postoperative pain in patients undergoing cesarean section, with no significant adverse effects. This study has substantial significance for clinical practice as it provides a safe and cost-effective pain management strategy for this population. However, additional research is required to validate these findings and determine the ideal dosage and administration time of melatonin.

## Figures and Tables

**Figure 1 medicina-59-01065-f001:**
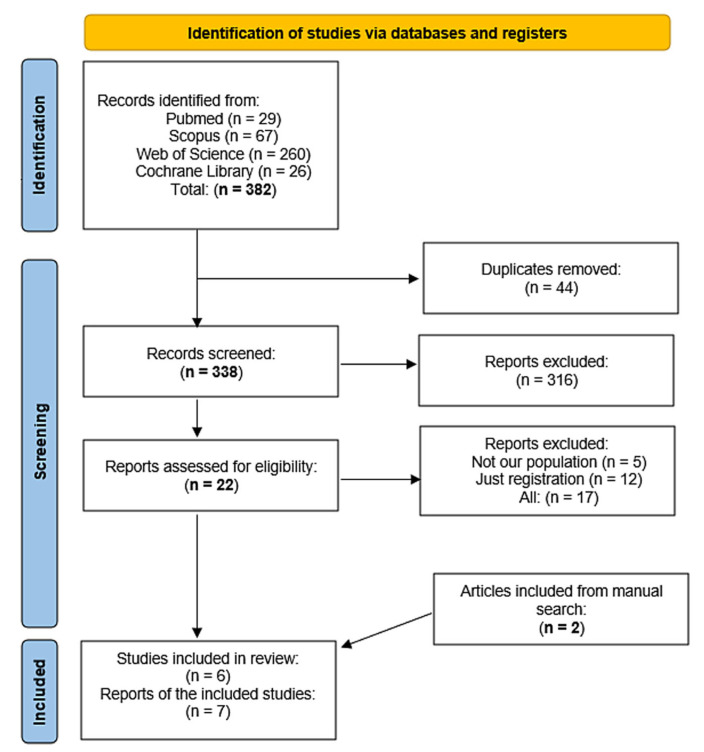
Depicts the PRISMA flowchart of the research selection procedure.

**Figure 2 medicina-59-01065-f002:**
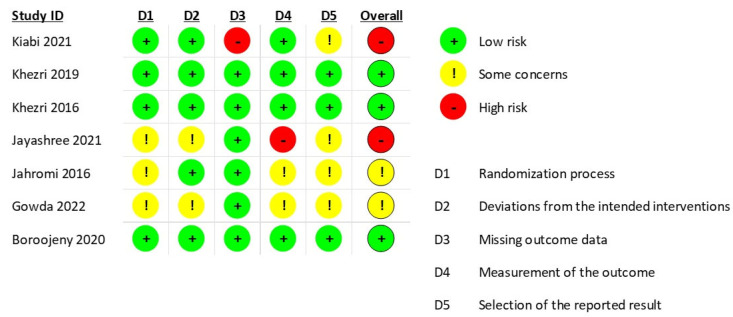
Depicts risk of bias summary of the included trials.

**Figure 3 medicina-59-01065-f003:**
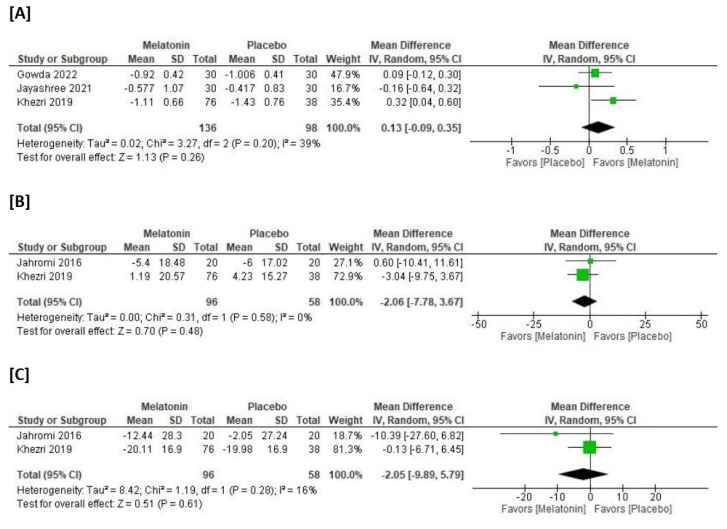
Meta-analysis of the mean change of (**A**) hemoglobin, (**B**) heart rate, and (**C**) mean arterial pressure.

**Figure 4 medicina-59-01065-f004:**
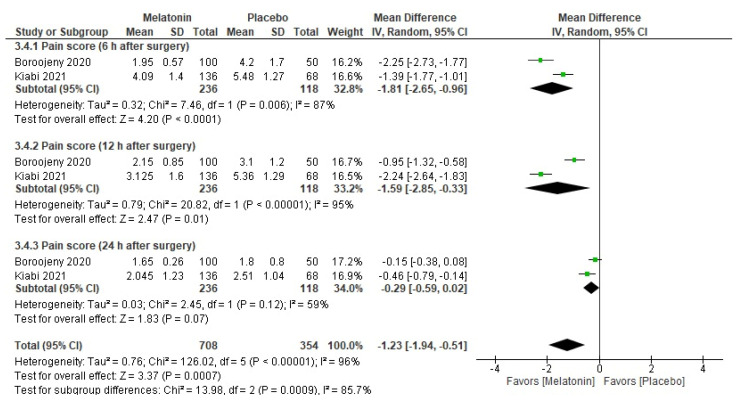
Meta-analysis of postoperative pain (10-point score).

**Figure 5 medicina-59-01065-f005:**
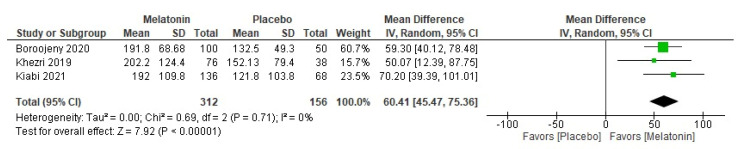
Meta-analysis of the time to first analgesia request.

**Figure 6 medicina-59-01065-f006:**
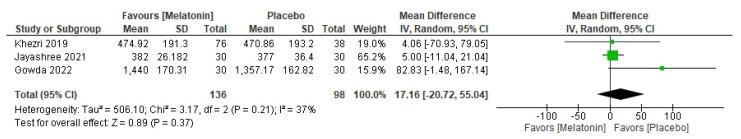
Meta-analysis of the total blood loss (mL).

**Figure 7 medicina-59-01065-f007:**
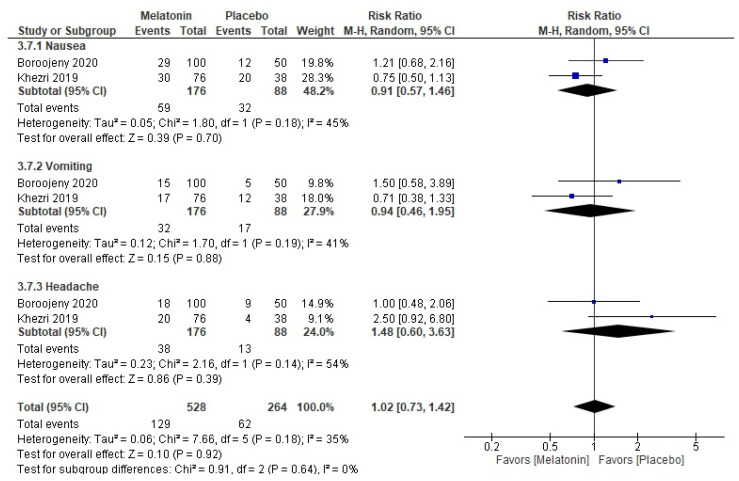
Meta-analysis of the rate of postoperative adverse events.

**Table 1 medicina-59-01065-t001:** Summary of the included trials.

Study ID	Design	Country	Interventions	Time	Dose	Type of Anesthesia	Administration Route	Sample Size	Follow-Up
Kiabi 2021 [30]	RCT	Iran	Melatonin	1 h before surgery	5 mg	Spinal anesthesia	Oral	68	24 h after surgery
					10 mg		Oral	68	
			Placebo		NA		Oral	68	
Khezri 2019 [29]	RCT	Iran	Melatonin	20 min before the spinal anesthesia	3 mg	Spinal anesthesia	Sublingual	40	12 h after surgery
					6 mg		Sublingual	40	
			Placebo		NA		Sublingual	40	
Khezri 2016 [16]	RCT	Iran	Melatonin	20 min before the spinal anesthesia	3 mg	Spinal anesthesia	Sublingual	40	12 h after surgery
					6 mg		Sublingual	40	
			Placebo		NA		Sublingual	40	
Jayashree 2021 [36]	RCT	India	Melatonin	20 min before the skin incision	6 mg	NR	Sublingual	30	NR
			Placebo		NA		Sublingual	30	
Jahromi 2016 [37]	RCT	Iran	Melatonin	1 h before spinal anesthesia	3 mg	Spinal anesthesia	Oral	20	NR
			Placebo		NA		Oral	20	
Gowda 2022 [38]	RCT	India	Melatonin	1 h before spinal anesthesia	3 mg	Spinal anesthesia	Oral	30	12 h after surgery
			Placebo		NA		Oral	30	
Boroojeny 2020 [39]	RCT	Iran	Melatonin	2 h before spinal anesthesia	1.5 mg	Spinal anesthesia	Oral	50	24 h after surgery
					3 mg		Oral	50	
			Placebo		NA		Oral	50	

RCT: randomized controlled trial; NA: not applicable; NR: not reported.

**Table 2 medicina-59-01065-t002:** Baseline characteristics of the population of the included trials.

Study ID	Interventions	Sample Size	Age Mean (SD)	Weight (kg) Mean (SD)	Duration of Surgery (min) Mean (SD)	Hemoglobin BCS (g/dL) Mean (SD)
Kiabi 2021 [30]	Melatonin	68	Between 40 and 20 years	NR	NR	NR
		68		NR	NR	NR
	Placebo	68		NR	NR	NR
Khezri 2019 [29]	Melatonin	40	28.19 (6.21)	74.75 (6.88)	81.70 (18.76)	12.83 (0.94)
		40	28.38 (5.67)	72.45 (6.59)	79.16 (20.11)	12.68 (0.75)
	Placebo	40	28.63 (5.31)	75.21 (7.15)	85.63 (15.70)	13.2 (0.8)
Khezri 2016 [16]	Melatonin	40	28.19 (6.21)	74.75 (6.88)	NR	NR
		40	28.38 (5.67)	72.45 (6.59)	NR	NR
	Placebo	40	28.63 (5.31)	75.21 (7.15)	NR	NR
Jayashree 2021 [36]	Melatonin	30	NR	NR	NR	11.02 (0.747)
	Placebo	30	NR	NR	NR	10.947 (0.564)
Jahromi 2016 [37]	Melatonin	20	30.21 (6.02)	NR	NR	NR
	Placebo	20	27.72 (4.01)	NR	NR	NR
Gowda 2022 [38]	Melatonin	30	NR	NR	NR	11.26 (0.87)
	Placebo	30	NR	NR	NR	11.82 (0.92)
Boroojeny 2020 [39]	Melatonin	50	26.2 (4.8)	NR	NR	NR
		50	26.5 (4.7)	NR	NR	NR
	Placebo	50	26.1 (4.8)	NR	NR	NR

NR: not reported.

**Table 3 medicina-59-01065-t003:** Subgroup analysis according to the melatonin doses.

Outcomes	M. 1.5 mg	M. 3 mg	M. 5 mg	M. 6 mg	M. 10 mg
Hemoglobin #	NR	0.12 g/dL, [−0.05, 0.30], *p* = 0.17	NR	0.16 g/dL, [−0.40, 0.71], *p* = 0.58	NR
Heart rate #	NR	−4.69 beat/min, [−14.40, 4.47], *p* = 0.30	NR	2.45 beat/min, [−5.21, 10.11], *p* = 0.53	NR
Mean arterial pressure #	NR	−2.21 mmHg, [−10.77, 6.35], *p* = 0.61	NR	−0.36 mmHg, [−7.85, 7.13], *p* = 0.93	NR
Pain score at 6 h #	−2.00 [−2.51, −1.49], *p* < 0.001	−2.50 [−2.97, −2.03], *p* < 0.001	−0.51 [−0.91, −0.11], *p* < 0.001	NR	−2.27 [−2.67, −1.87], *p* < 0.001
Pain score at 12 h #	−0.90 [−1.32, −0.48], *p* < 0.001	−1.00 [−1.40, −0.60], *p* < 0.00	−1.51 [−2.00, −1.02], *p* < 0.001	NR	−2.96 [−3.39, −2.53], *p* < 0.00
Pain score at 24 h #	−0.10 [−0.34, 0.14], *p* = 0.1	−0.20 [−0.43, 0.03], *p* = 0.9	0.08 [−0.31, 0.47], *p* = 0.68	NR	−1.01 [−1.30, −0.72], *p* < 0.001
Time to first analgesic request #	54.60 min [31.81, 77.39], *p* < 0.001	62.36 min [40.85, 83.87], *p* < 0.001	50.40 min [12.04, 88.76], *p* = 0.01	44.69 min [−2.13, 91.51], *p* = 0.06	90.00 min [57.25, 122.75], *p* < 0.001
Total blood loss #	NR	49.39 mL, [−19.17, 117.95], *p* = 0.16	NR	4.69 mL, [−11.07, 20.46], *p* = 0.56	NR
Nausea *	1.08 [0.55, 2.14], *p* = 0.82	0.93 [0.49, 1.79], *p* = 0.84	NR	0.81 [0.50, 1.29], *p* = 0.37	NR
Vomiting *	1.60 [0.56, 4.56], *p* = 0.38	0.76 [0.24, 2.36], *p* = 0.64	NR	0.95 [0.49, 1.85], *p* = 0.88	NR
Headache *	1.22 [0.56, 2.69], *p* = 0.62	0.71 [0.32, 1.57], *p* = 0.40	NR	4.28 [1.59, 11.48], *p* = 0.004	NR

# Mean difference; * risk ratio; NR: not reported.

## Data Availability

All data are available within the manuscript.

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
