# Peer review of "Efficacy and Safety of Preoperative Melatonin for Women Undergoing Cesarean Section: A Systematic Review and Meta-Analysis of Randomized Placebo-Controlled Trials"

_medicina, 2023, doi:10.3390/medicina59061065_

Round 1

Reviewer 1 Report

Good. However some figures need double-checking for accuracy as noted in my comments.

Many thanks of the opportunity to review the review article “Efficacy and Safety of Preoperative Melatonin for Women Undergoing Cesarean Section: A Systematic Review and Metaanalysis of Randomized Placebo-Controlled Trials”. The authors conducted a systematic review and meta-analysis on the efficacy of melatonin among patients undergoing CS. The authors concluded that preoperative melatonin is safe and can effectively decrease pain associated with CS postoperatively. Overall, the manuscript is well-written and follows the PRISMA guidelines. The topic is important to the special issues and within the scope. A major strength of the report is being the first-ever and this will highly increase citation. I have some minor comments before recommending the paper for acceptance.

1. Methods. For control group, please confirm if all control groups received placebo or some of them had no treatment (nothing).

2. Results. For the risk of bias assessment, please write up a paragraph and include description/justification of your judgments for each domain within the study. Details are missing.

3. Results. For Figure 3A and Figure 3B, please check the data for melatonin group for Khezri 2019 for accuracy. There seems no negative value compared with others.

4. Results. For Figure 4, please indicate if both studies used the same pain score scale (im assuming the 10-point VAS).

5. Results. For Figure 5, are you referring to opioids or NSAIDs?

6. Results. For Figure 6, why Gowada had an extremely high blood loss. Any reason for this particular study.

7. Discussion. The authors claimed that melatonin reduced substantially the amount of opioids, but there is no clear evidence for that. Are the authors referring to Figure about time to first analgesic? Because there is difference between time to first analgesic and amount of opioids used! Please rectify accordingly.

8. Discussion. The authors mentioned “We found a non-significant effect of melatonin on hemoglobin levels, heart rate, mean arterial pressure, or total blood loss, which can be explained by that there may be no direct influence of melatonin on these variables because heart rate and mean arterial pressure”. However, I am not pretty sure if this statement is completely true, since melatonin is sleep-inducer and it is anticipated to block sympathetic nervous system. Please cite relevant literature and revise accordingly. The lack of specific effects could be due to the small number of meta-analyzed studies (n=2).

9. Discussion. Please include additional limitations such as the heterogeneity of several outcomes and the wide differences in dosing between studies.

10. Please complement your research with evidence grading according to the GRADE system.

11. Highlight as a future direction the need for dose-response analysis of melatonin.

Minor editing of English language required

Author Response

Dear Editor-in-Chief, Associate Editor, and Peer-Reviewers,

We are very thankful to you for the pertinent notes; we have carefully read the comments and have revised the manuscript accordingly. Our responses are given in a point-by-point manner below. Additionally, all the changes applied to the manuscript are highlighted in yellow. We hope that in this new revised form, the manuscript will be suitable for publication in your esteemed journal. Thank you for taking time to review our paper and for your valuable suggestions. Overall, we believe that, thanks to your pertinent comments, the quality of the manuscript has been dramatically improved and the content is now more scientifically sound and intellectually solid.

Sincerely,

The Authors

Reviewer #1 report:

Reviewer (Q): 1. Methods. For control group, please confirm if all control groups received placebo or some of them had no treatment (nothing).

Author (A): Thank you for your comment. We made sure that all the included RCTs in the control group received placebo and none of the included RCTs received no treatment. 

Reviewer (Q): 2. Results. For the risk of bias assessment, please write up a paragraph and include description/justification of your judgments for each domain within the study. Details are missing.

Author (A): Thank you for your comment. We have added the following sentences in the RESULTS section: “Regarding randomization process domain, three RCTs [36-38] were evaluated as “some concerns” due to missing information about the randomization process. For deviation from intended interventions domain, two RCTs [36,38] were evaluated as “some concerns” due to no information were deviations from usual practice that were likely to impact on the outcome. However, for missing data domain, one RCT [30] were evaluated as “high” risk of bias due to a high degree of missing data. However, for measurement of outcomes domains, two RCTs [37,38] were evaluated as some concerns and one RCT [36] were evaluated as “high” risk of bias. Also, for selection of the reported results domain, four RCTs [30, 36-38] were evaluated as “some concerns” due to no information available to exclude the possibility that reported outcome data were selected.”

Reviewer (Q): 3. Results. For Figure 3A and Figure 3B, please check the data for melatonin group for Khezri 2019 for accuracy. There seems no negative value compared with others.

Author (A): Thank you for your comment. We checked all the values and it was reported correctly.

Reviewer (Q): 4. Results. For Figure 4, please indicate if both studies used the same pain score scale (im assuming the 10-point VAS).

Author (A): Thank you for your comment. Both studies were used 10-point rating scale (Kiabi 2021 “VAS”, and Boroojeny 2020 “NRS”).

Reviewer (Q): 5. Results. For Figure 5, are you referring to opioids or NSAIDs?

Author (A): Thank you for your comment. It depends on the study protocol, none of the included studies specify whether opioids or NSAIDS.

Reviewer (Q): 6. Results. For Figure 6, why Gowada had an extremely high blood loss. Any reason for this particular study.

Author (A): Thank you for your comment. Unfortunately, in this study; most of the required information were missed, thus this study was evaluated as “some concerns” risk of bias.

Reviewer (Q): 7. Discussion. The authors claimed that melatonin reduced substantially the amount of opioids, but there is no clear evidence for that. Are the authors referring to Figure about time to first analgesic? Because there is difference between time to first analgesic and amount of opioids used! Please rectify accordingly.

Author (A): Thank you for your comment. Sorry for this mistake, we remove this sentence from DISCUSSION section.

Reviewer (Q): 8. Discussion. The authors mentioned “We found a non-significant effect of melatonin on hemoglobin levels, heart rate, mean arterial pressure, or total blood loss, which can be explained by that there may be no direct influence of melatonin on these variables because heart rate and mean arterial pressure”. However, I am not pretty sure if this statement is completely true, since melatonin is sleep-inducer and it is anticipated to block sympathetic nervous system. Please cite relevant literature and revise accordingly. The lack of specific effects could be due to the small number of meta-analyzed studies (n=2).

Author (A): Thank you for your comment. We added the following sentences into DISCUSSION section “However, the association between melatonin and heart rate and mean arterial pressure could be explained by the effect of melatonin in sympathetic nervous system [18].”

Reviewer (Q): 9. Discussion. Please include additional limitations such as the heterogeneity of several outcomes and the wide differences in dosing between studies.

Author (A): Thank you for your comment. We have added this information into LIMITATION section.

Reviewer (Q): 10. Please complement your research with evidence grading according to the GRADE system.

Author (A): Thank you for your comment. We have added a GRADE assessment into Table S2. 

Reviewer (Q): 11. Highlight as a future direction the need for dose-response analysis of melatonin.

Author (A): Thank you for your comment. We have added the following sentence in the DISCUSSION section “Also, further secondary analysis like dose-response analysis are needed to determine the ideal dose of melatonin administration.”

Reviewer 2 Report

1. Some sections of methods need reference (s) such as the Der Simonian-Liard meta-analysis model, I-square, ...

2. Subgroup analysis is unclear (Table 3).

3. Why did not o you perform SMD?

There are several grammatical errors.

Thereare several grammatical error.

Author Response

Response to Peer-Reviewers

Dear Editor-in-Chief, Associate Editor, and Peer-Reviewers,

We are very thankful to you for the pertinent notes; we have carefully read the comments and have revised the manuscript accordingly. Our responses are given in a point-by-point manner below. Additionally, all the changes applied to the manuscript are highlighted in yellow. We hope that in this new revised form, the manuscript will be suitable for publication in your esteemed journal. Thank you for taking time to review our paper and for your valuable suggestions. Overall, we believe that, thanks to your pertinent comments, the quality of the manuscript has been dramatically improved and the content is now more scientifically sound and intellectually solid.

Sincerely,

The Authors

Reviewer (Q): 1. Some sections of methods need reference (s) such as the Der Simonian-Liard meta-analysis model, I-square, ...

Author (A): Thank you for your comment. We cited all of these models with reference #32.

Reviewer (Q): 2. Subgroup analysis is unclear (Table 3).

Author (A): Thank you for your comment. We added supplemental figures for subgroup analysis for more clarification.

Reviewer (Q): 3. Why did not o you perform SMD?

Author (A): Thank you for your comment. All of the included RCTs have the same measurement tools, therefore; we used MD instead of SMD.

Reviewer (Q): There are several grammatical errors.

Author (A): Thank you for your comment. Our manuscript has been approved by native English speaker.

Reviewer 3 Report

A well presented and comprehensive manuscript on the safety and efficacy of preoperative melatonin on postoperative outcomes on pregnant women undergoing elective cesarean section.

The investigation confirmed that administration of melatonin before surgery is safe and effective in lowering postoperative pain in cesarean patients and had no significant side effects.

The approach is scientifically sound and research methods are appropriate and adequate. Results are well presented and are in line with the conclusions furthermore, the figures and tables in the text are appropriate and easy to interpret. The references cited are relevant and most of the publications are recent.

I particularly appreciate the practical implications of this study in reassuring health workers on the safety of using melatonin in cesarean patients. I nonetheless agree with the authors that the number of included studies and the  sample size were rather limited and therefore merit additional research to validate the findings and determine ideal dosage and timing of administering melatonin.

To conclude, I recommend that the manuscript be accepted for publication, remember to incorporate the few minor corrections indicated in the text.

The quality of English is fine, requires only minor editing some of it is indicated in the text.

Author Response

Response to Peer-Reviewers

Dear Editor-in-Chief, Associate Editor, and Peer-Reviewers,

We are very thankful to you for the pertinent notes; we have carefully read the comments and have revised the manuscript accordingly. Our responses are given in a point-by-point manner below. Additionally, all the changes applied to the manuscript are highlighted in yellow. We hope that in this new revised form, the manuscript will be suitable for publication in your esteemed journal. Thank you for taking time to review our paper and for your valuable suggestions. Overall, we believe that, thanks to your pertinent comments, the quality of the manuscript has been dramatically improved and the content is now more scientifically sound and intellectually solid.

Sincerely,

The Authors

Reviewer (Q): The quality of English is fine, requires only minor editing some of it is indicated in the text.

Author (A): Thank you for your comment. We Adjusted these mistakes according to your suggestion.